# Transforming Psoriasis Care: Probiotics and Prebiotics as Novel Therapeutic Approaches

**DOI:** 10.3390/ijms241311225

**Published:** 2023-07-07

**Authors:** Mihaela Cristina Buhaș, Rareș Candrea, Laura Ioana Gavrilaș, Doina Miere, Alexandru Tătaru, Andreea Boca, Adrian Cătinean

**Affiliations:** 1Department of Dermatology, Toxicology and Clinical Pharmacology, “Iuliu Hatieganu” University of Medicine and Pharmacy, 400423 Cluj-Napoca, Romania; drmihaelabuhas@gmail.com (M.C.B.); dr.tataru@yahoo.com (A.T.); 2Master Program in Nutrition and Quality of Life, “Iuliu Hatieganu” University of Medicine and Pharmacy, 400423 Cluj-Napoca, Romania; raresadelin99@gmail.com; 3Department of Bromatology, Hygiene, Nutrition, “Iuliu Hatieganu” University of Medicine and Pharmacy, 23 Marinescu Street, 400337 Cluj-Napoca, Romania; dmiere@umfcluj.ro; 4Department of Pharmacology, Toxicology and Clinical Pharmacology, “Iuliu Hatieganu” University of Medicine and Pharmacy, 400337 Cluj-Napoca, Romania; boca.andreea@umfcluj.ro; 5Department of Internal Medicine, Faculty of Medicine, “Iuliu Hatieganu” University of Medicine and Phamacy, 400423 Cluj-Napoca, Romania; catinean@gmail.com

**Keywords:** psoriasis, probiotics, prebiotics, gut microbiota, inflammation

## Abstract

Psoriasis is a chronic inflammatory skin disease with autoimmune pathological characteristics. Recent research has found a link between psoriasis, inflammation, and gut microbiota dysbiosis, and that probiotics and prebiotics provide benefits to patients. This 12-week open-label, single-center clinical trial evaluated the efficacy of probiotics (*Bacillus indicus* (HU36), *Bacillus subtilis* (HU58), *Bacillus coagulans* (SC208), *Bacillus licheniformis* (SL307), and *Bacillus clausii* (SC109)) and precision prebiotics (fructooligosaccharides, xylooligosaccharides, and galactooligosaccharides) in patients with psoriasis receiving topical therapy, with an emphasis on potential metabolic, immunological, and gut microbiota changes. In total, 63 patients were evaluated, with the first 42 enrolled patients assigned to the intervention group and the next 21 assigned to the control group (2:1 ratio; non-randomized). There were between-group differences in several patient characteristics at baseline, including age, psoriasis severity (the incidence of severe psoriasis was greater in the intervention group than in the control group), the presence of nail psoriasis, and psoriatic arthritis, though it is not clear whether or how these differences may have affected the study findings. Patients with psoriasis receiving anti-psoriatic local therapy and probiotic and prebiotic supplementation performed better in measures of disease activity, including Psoriasis Area and Severity Index, Dermatology Life Quality Index, inflammatory markers, and skin thickness compared with those not receiving supplementation. Furthermore, in the 15/42 patients in the intervention group who received gut microbiota analysis, the gut microbiota changed favorably following 12 weeks of probiotic and prebiotic supplementation, with a shift towards an anti-inflammatory profile.

## 1. Introduction

Psoriasis is a chronic inflammatory skin disease with a significant hereditary predisposition and autoimmune pathological characteristics [1]. Psoriasis is more prevalent in high-income countries and occurs more frequently in adults, compared to children [2]. Psoriatic arthritis is the most well known complication of this disease, although it is also associated with an increased risk of cardiovascular disease, diabetes mellitus, obesity, and hyperlipidemia [3].

The pathogenesis of psoriasis is characterized by uncontrolled proliferation and dysfunctional differentiation of keratinocytes. Histologically, psoriatic plaques show epidermal hyperplasia overlying inflammatory infiltrates of dermal dendritic cells, macrophages, T cells, and neutrophils. Tumor necrosis factor-α (TNF-α), interferon-γ (IFN-γ), interleukin (IL)-23/IL-17A, and IL-22 are the primary immunological molecules responsible for keratinocyte proliferation, inflammation, and cytokine infiltration in psoriasis [1,4]. Previous studies have shown the importance of inflammatory cytokines in the pathogenesis of psoriasis, as these patients present with higher levels of circulating cytokines compared to healthy controls [5,6]. Furthermore, there is increasing evidence showing that the gut microbiota plays a critical role in autoimmune diseases, including psoriasis [7,8]. Recent studies have shown that patients with psoriasis present with gut microbiota dysbiosis, along with an abnormal immune response due to elevated inflammatory molecules [9,10,11].

Probiotic supplementation could represent a novel therapeutic strategy in the treatment of gut microbiota dysbiosis in patients with psoriasis, as supplementation has been associated with significant alleviation of psoriasis-like pathogenic characteristics and a reduced proinflammatory status, in both experimental and clinical trials [12,13,14]. Spore-forming *Bacillus* probiotics have several advantages over non-spore-forming probiotics, as they can tolerate room temperature storage and are more resistant to stomach acids [15]. As such, they survive the harsh conditions of gastric passage, entering into the intestines completely viable, where they are able to germinate in high numbers [16,17]. Supplementation with spore-based *Bacillus* probiotics has been shown to improve markers of gut health [18,19,20]. More specifically, supplementation with a probiotic containing five *Bacillus* probiotic strains (*Bacillus indicus* HU36, *Bacillus subtilis* HU58, *Bacillus coagulans* SC208, *Bacillus licheniformis* SL307, and *Bacillus clausii* SC109) resulted in increased gut microbiome diversity, increases in bacterial species considered beneficial to the human host, increased levels of short-chain fatty acids, and a reduction in inflammatory markers [20,21]. Each strain included in this probiotic mix has unique benefits. For example, *B. licheniformis* produces proteases that aid in protein digestion [22], and B vitamins [23] which are a critical component of human health; *B. indicus* produces carotenoids [24,25] which are potent antioxidants that reduce inflammation; *B. subtilis* produces vitamin K2 [26] which is important for skin, bone, brain, and heart health; *B. clausii* supports both gut barrier and immune function [27]; and *B. coagulans* supports immune system function [28] and produces L-lactic acid [29]. This probiotic mixture has also been evaluated in combination with a prebiotic oligosaccharide mixture (fructooligosaccharides, xylooligosaccharides, and galactooligosaccharides) [30]. In that study, combination treatment resulted in stronger stimulation of short-chain fatty acid production compared with a similar study of probiotic supplementation alone, as well as increases in *Facealibacterium prausnitzii* and *Lactobacillus* spp. that were not observed with probiotic supplementation alone, indicating a potential advantage for co-supplementation [30]. Fructooligosaccharides, xylooligosaccharides, and galactooligosaccharies are reported to improve intestinal barrier function and to have anti-inflammatory, immunomodulatory, and bifidogenic effects [31,32,33]. Considering the inflammatory status and gut microbiota dysbiosis in psoriasis and the fact that, to the authors’ knowledge, spore-based probiotics have not been evaluated in patients with psoriasis, the aim of this study was to assess the efficacy of a spore-based probiotic combined with a precision prebiotic supplementation in patients with psoriasis receiving topical therapy, with an emphasis on potential metabolic, immunological, and gut microbiota changes.

## 2. Results and Discussion

### 2.1. Patients and Background Characteristics

In total, 63 patients with psoriasis were enrolled in this study: 42 in the intervention group and 21 in the control group.

Table 1 shows a comparison of background characteristics between the intervention and control groups. There were no statistically significant differences in sex distribution, smoking status, family history of psoriasis, and inflammatory bowel disease history between the groups (*p* < 0.05). There was a significant between-group difference in age, with a greater mean age in the control group than in the intervention group (*p* < 0.001). Additionally, significant between-group differences were found for psoriasis severity (*p* = 0.006), the presence of nail psoriasis (*p* = 0.028), and psoriatic arthritis (*p* < 0.001), with more patients in the intervention group presenting with severe psoriasis status (Psoriasis Area and Severity Index (PASI) > 10) and psoriasis complications. This could explain why the proportion of patients with previous corticosteroid (i.e., corticosteroid use prior to 2 months prior to study entry) use was significantly higher in the intervention group than in the control group (*p* = 0.016).

### 2.2. PASI, DLQI, and Anthropometric Measures

As presented in Table 2, following 12 weeks of probiotic and prebiotic administration, PASI and Dermatology Life Quality Index (DLQI) scores significantly decreased in the intervention group (*p* < 0.001), compared to the control group. PASI and DLQI scores are markers that reliably correlate with psoriasis severity and quality of life, respectively, for patients with psoriasis [34]. In line with our results, recent clinical trials have highlighted that probiotic and/or prebiotic supplementation in patients with psoriasis resulted in significantly lower PASI and DLQI scores, indicating improvement in disease severity and quality of life [13,35,36]. As psoriasis has a significant negative impact on health-related quality of life [37], our findings suggest that probiotic/prebiotic supplementation could be beneficial in the management of the disease, along with conventional anti-psoriatic treatment.

Although we observed no significant difference between the two groups in terms of fat mass (FM) changes (*p* > 0.05), curiously, a significant difference was found regarding body mass index (BMI) and free fat mass (FFM) distribution (*p* < 0.05), with the mean ± standard deviation (SD) for both decreasing in the intervention group compared to the control group (BMI, 24.50 ± 7.85 versus 24.45 ± 8.4; FFM, 52.61 ± 10.252 versus 50.254 ± 10.198; *p* < 0.05) (Table 3). As part of a healthy lifestyle, probiotic and prebiotic supplementation has previously been addressed in the management of obesity, with supplementation being positively correlated with BMI reduction [38]. Patients in the intervention group had a weight reduction after 12 weeks of supplementation, which was caused by the decrease in FFM rather than a loss of FM. These variations might be attributed to interindividual differences related to lifestyle factors, such as physical activity level and dietary intake.

### 2.3. Inflammatory Markers

The effects of probiotic and prebiotic supplementation on inflammatory markers are presented in Table 4. There were substantial interindividual variations in baseline cytokine levels, specifically for TNF-α, IL-6, and IL-17A. Several factors may contribute to such variation in patients with psoriasis. First, psoriasis is a heterogeneous disease that exhibits diverse clinical manifestations and has a range of disease severity. This inherent variation may influence immune responses and, consequently, cytokine production. Additionally, genetic and environmental factors, such as lifestyle, diet, and comorbidities, may influence cytokine production. Such interindividual variations in inflammatory markers, both in healthy individuals and those with psoriasis, are commonly reported in the literature [39,40,41,42,43]. After 12 weeks of supplementation, TNF-α, IL-6, IFN-γ, and IL-10 levels varied significantly between the two groups (*p* < 0.05). Mean ± SD TNF-α significantly decreased in the intervention group from 50.025 ± 41.40 to 15.11 ± 36.21 (*p* < 0.05) and numerically increased in the control group (baseline to Week 12). Although the between-group differences were not significant, mean ± SD IL-6 and IFN-γ levels were numerically decreased in the intervention group and numerically increased in the control group. There was a significant increase in IL-10 in the intervention group (*p* < 0.001) and a significant decrease in the control group (*p* < 0.05). IL-10 is recognized as an essential immunoregulatory cytokine due to its anti-inflammatory characteristics. Induction of IL-10 expression was discovered to be a major suppressor of cellular immune response in psoriasis, and the benefits of IL-10 as a biological treatment are still being explored in clinical studies [44]. Furthermore, we found a significant increase in IL-17A levels in the control group (*p* < 0.05), though there were no significant between-group differences (*p* > 0.05). We note that reference levels for cytokines in patients with psoriasis are not yet established. Therefore, cytokine levels are often interpreted in conjunction with clinical symptoms and other diagnostic measures. In clinical practice, it is common to compare an individual’s cytokine levels to their baseline level, as longitudinal trends can provide meaningful clinical insights.

The association between probiotic supplementation in patients with psoriasis and changes in inflammatory markers is yet an active research field, but preliminary findings have revealed promising effects [8,45]. In psoriatic mice, probiotic supplementation was previously linked with an improvement in psoriasis symptoms, as well as a decrease in skin inflammatory markers, such as TNF-α, IL-6, and IL-17A [46]. One mechanism explaining the favorable effects of probiotic supplementation in patients with psoriasis may be related to increased gut levels of acetate and propionate, which have been linked to the suppression of cytokine activity through the IL-23/T helper cells (Th)17 axis [14]. Moludi et al. showed that supplementation of *Lactobacillus* stains in 50 patients with psoriasis for 8 weeks resulted in an anti-inflammatory status, demonstrated by an increase in total antioxidant capacity and a decrease in C-reactive protein (CRP), IL-6, and malondialdehyde levels [35]. Along the same line, Groeger et al. highlighted that supplementation with *Bifidobacteria infantis* for 6–8 weeks reduced levels of CRP and TNF-α in patients with psoriasis [47]. 

### 2.4. Metabolic Parameters

After adjusting the cholesterol level results for sex, age, and baseline values, our results showed a significant difference between groups in total cholesterol (C), low-density lipoprotein (LDL)-C, and high-density lipoprotein (HDL)-C (*p*-adjusted < 0.05). Although the difference in total cholesterol levels between baseline and Week 12 was not significant in the intervention group (*p* > 0.05), LDL-C and HDL-C values were considerably improved in the intervention group compared to the control group (*p* < 0.001 and *p* < 0.05, respectively). Moreover, significant between-group differences in triglyceride levels were also observed (*p* = 0.024) (Table 5). Prebiotic supplements, such as inulin and fructooligosaccharides, have demonstrated efficacy in lowering cholesterol and triglyceride levels in both in vivo and in vitro studies [48]. Moreover, a systematic review of clinical trials assessing the effects of probiotic supplementation on hypercholesterolemic participants showed that *Lactobacillus acidophilus*, *Bifidobacterium longum*, and *Lactobacillus plantarum* may have hypocholesterolemic properties, although further studies with various mixtures and dosages of probiotics in a greater number of volunteers are needed [49]. According to previous studies, patients with psoriasis had higher plasma cholesterol levels compared to non-psoriatic individuals [50]; therefore, reducing lipid levels through lifestyle changes and administration of supplements, including probiotics and prebiotics, may be beneficial.

Our study demonstrated that probiotic and prebiotic supplementation for 12 weeks resulted in a significant decrease in uric acid (*p* = 0.001). Previous studies have shown that patients with psoriasis, especially those with severe disease, had a higher risk of hyperuricemia compared to non-psoriatic controls [51,52]. Additionally, hyperuricemic individuals with psoriatic arthritis had a worse response to anti-psoriatic medication and presented with more joint destruction compared with those who were normouricemic [53]. 

In contrast to findings that showed an improvement in metabolic syndrome markers after probiotic supplementation [54,55], our study found that blood sugar and insulin levels were considerably higher in the intervention group following probiotic and prebiotic administration (*p* = 0.001 for both). 

### 2.5. Ultrasound Assessments

Using the data generated from the high-frequency ultrasonography of psoriatic skin, we found significant differences in psoriasis plaque hydration, perilesional area hydration, and subepidermal low-echogenic band (SLEB) between the two groups (*p* < 0.05), with both the control and intervention groups showing remarkable improvements (*p* < 0.05). Regarding skin thickness, only the intervention group showed a significant reduction from baseline to end of study (mean ± SD change, 237.5 ± 276; *p* < 0.001) (Table 6). The changes in ultrasound markers may reflect the efficacy of the topical therapy administered during the study. Although SLEB is not specific to psoriasis, it is a strong marker of collagen degradation and inflammation [56]. When compared to healthy skin, psoriasis plaques showed an increase in skin thickness, caused by underlying inflammation, edema, and vascularity variations [57,58]. In agreement with previous research that found a beneficial relationship between oral probiotics and a decrease in ultraviolet-caused epidermal thickness [59], our study showed that probiotic and prebiotic supplementation contributed to the reduction in skin thickness in the intervention group.

### 2.6. Gut Microbiota Changes

After 12 weeks of probiotic and prebiotic supplementation, we identified 12 markers that differed significantly from baseline, including changes in seven bacterial abundances (Figure 1). Shannon index (mean ± SD, −0.169 ± 0.222), *Firmicutes*/*Bacteroidetes* ratio (−0.413 ± 0.294), and acetate/propionate production (–4.686 ± 6.531) significantly increased from baseline to Week 12 in the intervention group (*p* < 0.05), and the *Prevotella*/*Bacteroidetes* ratio (0.920 ± 1.425) and number of lipopolysaccharide (LPS)-positive bacteria (1.044 ± 1.531) significantly decreased (*p* < 0.05) (Table 7). Regarding specific abundances, our results showed a significant increase in *Ruminococcus* spp., *Verrucomicrobia*, and *Akkermansia muciniphila* (*p* < 0.05), while *Bacteroidetes*, *Prevotella* spp., *Prevotella copri*, and *Clostridium difficile* significantly decreased (*p* < 0.05) in the intervention group from baseline to Week 12 (Table 8). 

The Shannon index is a parameter used to assess gut microbiota bacterial diversity, with higher values signifying greater community diversity [60,61]. As previously observed, probiotic supplementation is an important modulator for preventing a low bacterial diversity of gut microbiota in psoriatic mice [14]. However, in patients with psoriasis, probiotic administration for 12 weeks did not result in significant changes in gut microbiota bacterial diversity, although the Shannon index was decreased to values comparable to those in the healthy population [62]. Despite the dysbiosis identified in the gut microbiota of patients with psoriasis, the relationship between beneficial and harmful bacteria taxa is far more important in the characterization of the psoriatic intestinal microbiota, compared to the overall diversity [63]. 

Because of its importance in maintaining intestinal homeostasis, the *Firmicutes*/*Bacteroidetes* ratio has been given considerable attention in recent years. Alterations in the *Firmicutes*/*Bacteroidetes* ratio have been associated with gut dysbiosis, obesity, and intestinal inflammatory diseases, while increasing the *Firmicutes*/*Bacteroidetes* ratio using probiotic supplements has been associated with a protective and anti-inflammatory intestinal effect [64]. Precisely, in the gut microbiota, a higher *Firmicutes*/*Bacteroidetes* ratio has been linked to increased acetate synthesis, with both acetate and propionate being recognized as modulating not only gut-specific but also distant inflammatory responses via the IL-23/Th17 axis [14,65]. Moreover, Groeger et al. showed that *Bifidobacteria infantis* supplementation reduced the proinflammatory status by reducing LPS-stimulated TNF-α and IL-6 levels in healthy participants [47]. This might explain the reduction in LPS-positive bacteria, along with the alleviation of psoriasis symptoms and the decrease in inflammatory cytokines, such as TNF-α and IL-6, found in our study.

Gut microbiota dysbiosis, which is common in patients with psoriasis [9], may induce an abnormal immune response [10]. Studies characterizing the intestinal microbiota of patients with psoriasis showed conflicting results regarding taxa distribution, making it difficult to develop a broad picture of the psoriatic intestinal microbiota [9,11,66]. However, a reduction in *Akkermansia muciniphila* and *Ruminococcus* spp. was previously observed in patients with psoriasis [11,67]. *Akkermansia muciniphila* and *Ruminococcus* spp. are mucin-degrading bacteria and are associated with propionate and acetate production and intestinal mucosa integrity [68,69].

### 2.7. Limitations

This study had several limitations that should be considered when interpreting the results. First, due to the relatively small sample size (*n* = 63), the results are not considered to be representative of the entire population of patients with psoriasis. Second, the risk of bias was higher in our study due to the methodology used, given that the study was not randomized or double-blinded. Third, there were significant differences at baseline between the intervention and control groups for several parameters, including severity of psoriasis, presence of nail psoriasis, presence of psoriatic arthritis, and some inflammatory markers (TNFα, IL-17A, and IL-6). It is unclear how these between-group differences may have impacted the study findings. Fourth, gut microbiota analysis was only conducted in 15/42 patients in the intervention group and none of the patients in the control group; therefore, we were not able to compare gut microbiota changes between the two groups or to have a complete dataset of changes for all 42 patients in the intervention group. Fifth, we are not able to decipher whether each of the changes was a result of probiotic supplementation, prebiotic supplementation, or the cumulative effects of both, given that we did not have probiotic-only and prebiotic-only groups. Finally, we did not investigate lifestyle factors, such as dietary intake or physical activity level, which are known to modulate gut microbiota diversity, metabolic markers, and anthropometric data.

## 3. Materials and Methods

### 3.1. Ethical Considerations

This study was conducted according to the guidelines outlined in the Declaration of Helsinki, the Amsterdam Protocol, and Directive 86/609/EEC. The study protocol was approved by the Ethical Commission of the “Iuliu Haţieganu” University of Medicine and Pharmacy Cluj-Napoca (No. 267/06.30.2021). Prior to enrollment, all patients provided written informed consent to participate in the study.

### 3.2. Study Design and Population

Figure 2 highlights the design and flow of the study. This 12-week open-label, single-center clinical trial included patients with psoriasis who were treated in a dermatological private practice between October 2021 and September 2022. Patients who met the following criteria were included in the study: age ≥ 18 years, clinical and histopathological diagnosis of psoriasis vulgaris with active psoriatic plaques on the skin, no current anti-psoriatic treatment, and no diagnosis of other associated diseases. The exclusion criteria were as follows: presence of inflammatory or autoimmune diseases other than psoriasis; previous administration of probiotic supplements in the month before the first study screening; previous administration of antibiotics 6 weeks before trial entry; previous administration of oral corticosteroids 2 months before trial entry; methotrexate, cyclosporine, or biologic treatment in the past 3 months; history of cancer, except for basal or squamous cell carcinoma of the skin that had been treated and was considered cured for at least 3 years prior to study entry; any allergies to the study probiotic and prebiotic supplements or any of the ingredients in the capsules. Subjects who were pregnant or breastfeeding and those who planned to become pregnant during the study period were not enrolled in the study. All participants agreed to use a highly effective contraceptive method throughout the study. Further, patients who were expected to have a high sun exposure during the study, who had signs of bacterial infection, who presented with skin conditions other than psoriasis prior to trial enrollment, and who had participated in other clinical trials in the 3 months prior to our study were not recruited. 

Enrolled patients were allocated at a 2:1 ratio to the intervention and control groups. The first 42 subjects enrolled in the study were assigned to the intervention group, and the next 21 were allocated to the control. Both groups received local anti-psoriatic treatment for 12 weeks as follows: betamethasone cream 0.5% (weeks 1–4, daily; weeks 4–8, 3 times per week; weeks 8–12, once per week) and 10% urea cream applied twice daily. Besides the topical treatment, the intervention group received 2 probiotic capsules daily (MegaSporeBiotic, Microbiome Labs, Glenview, IL, US) with their mid-day meal for 12 weeks. Each capsule contained 2 billion (2 × 10^9^) colony-forming units (CFUs) of a mixture of five strains of *Bacillus* spores (*B. indicus* (HU36), *B. subtilis* (HU58), *B. coagulans* (SC208), *B. licheniformis* (SL307), and *B. clausii* (SC109)). Additionally, patients in the intervention group received 5 g daily of MegaPre (Microbiome Labs), a precision prebiotic oligosaccharide mixture (fructooligosaccharides, xylooligosaccharides, and galactooligosaccharides) dissolved in 450 mL of water, which was taken on an empty stomach for the last 8 weeks of the 12-week study. The control group only received anti-psoriatic topical treatment. Following trial enrollment, patients received instructions regarding the use of local treatment and the administration of probiotic and prebiotic supplements.

### 3.3. Data Collection

Background demographic and psoriasis diagnosis data were collected at the screening visit. Blood was collected at the beginning (screening visit) and end (Week 12) of the study. Levels of serum TNF-α, IL-6, IFN-γ, IL-17A, IL-10, and insulin were determined using commercially available enzyme-linked immunosorbent assay kits according to the manufacturer’s protocol (Elabscience, Houston, TX, USA). Total cholesterol, triglyceride, HDL-C, blood glucose, and uric acid levels were analyzed using an Indiko Plus Chemistry Analyzer (Thermo Fisher Scientific, Waltham, MA, USA). For the LDL-C determination, the following formula was used: ([total cholesterol − HDL] − triglyceride)/5. A previously described high-frequency ultrasonography technique was used for skin examinations [70]. For the examination of psoriasis plaques and perilesional area hydration, the water-holding capacity in the stratum corneum was determined at the beginning and end of the study. Using the DermaLab Skin analysis technology, but with the addition of a hydration pin probe, we performed three measurements for each area of the skin. The mean of those three measurements was reported. Anthropometric data were collected at the screening visit and at Week 12, including weight, height, BMI, FM, and FFM. A TANITA Body Composition Analyzer SC-240MA (TANITA, Tokyo, Japan) was used to determine weight, BMI, FM, and FFM. The PASI and the DLQI were evaluated at the screening visit and at Week 12. Moreover, at the screening visit and Week 12, gut microbiota analysis was conducted for 15 patients from the intervention group. Fecal samples were collected and sent to an external laboratory (Bioclinica-Romania in collaboration with Ganzimmun Diagnostics AG-Germany) for processing. Gut microbiota analysis was conducted using next-generation sequencing technology [71,72].

### 3.4. Statistical Analysis

For normally distributed data, the mean and SD are reported; median and IQR are reported for non-normally distributed data. The paired-samples t-test and related-samples Wilcoxon signed rank test were used to detect differences among patients within the same group (i.e., from baseline to Week 12). Data from the intervention and control groups were compared using Pearson’s chi-square test. Between-group differences were analyzed using the independent-samples t-test and the independent-samples median test. An analysis of covariance (ANCOVA) test, adjusted for age, sex, and the baseline value of the parameter being evaluated, was used to compare the post-intervention final results. A *p*-value < 0.05 was considered statistically significant. Statistical Package for Social Sciences version 20.0.0 software (SPSS Inc., Chicago, IL, USA) was used for all statistical analyses.

## 4. Conclusions

According to our findings, probiotic and prebiotic supplementation enhanced the general health of patients with psoriasis who were receiving local anti-psoriatic treatment. First, probiotic and prebiotic supplementation significantly improved quality of life, as demonstrated by a reduction in PASI and DLQI scores. Second, the BMI and FFM of patients in the intervention group were significantly reduced (end of trial versus baseline). Third, prebiotic and probiotic (*Bacillus* spp.) supplementation promoted an anti-inflammatory response and contributed to improved symptoms in the intervention group by regulating cytokine activity (reduced levels of TNFα, IL-6, and IFN-γ, and enhanced levels of IL-10). Also, at the end of the study, the intervention group had significantly improved metabolic markers compared to the control group, including lower levels of total cholesterol, LDL-C, triglyceride, and uric acid; HDL-C levels increased. Finally, supplementation enhanced the diversity of gut microbiota (shown by a higher Shannon index), increased the *Firmicutes*/*Bacteroidetes* ratio and acetate/propionate production, and reduced the *Prevotella*/*Bacteroidetes* ratio and the number of LPS-positive bacteria. Moreover, there was a reduction in *Bacteroidetes* spp., *Prevotella* spp., *Prevotella copri*, and *Clostridium difficile*, while *Akkermansia muciniphila*, *Verrucomicrobia*, and *Ruminococcus* spp. levels increased (*p* < 0.05).

Overall, our study showed that spore-based probiotic and precision prebiotic supplementation in patients with psoriasis who were receiving adequate anti-psoriatic therapy experienced multiple improvements, including improvements in quality of life, inflammation, and gut microbiota dysbiosis. However, additional clinical trials are needed to fully elucidate the potential benefits of probiotics and prebiotics for psoriasis and to identify the most effective combination and dose. 

## Figures and Tables

**Figure 1 ijms-24-11225-f001:**
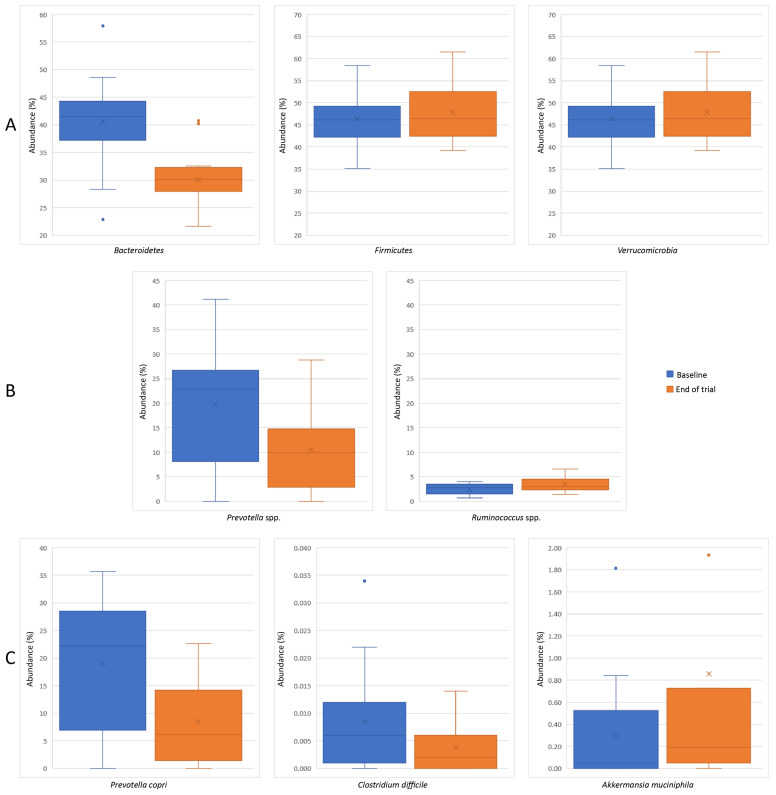
Abundances that showed a significant difference (*p* < 0.05) from baseline to Week 12 of probiotic and prebiotic administration ((**A**) phylum level, (**B**) species level, (**C**) bacteria level).

**Figure 2 ijms-24-11225-f002:**
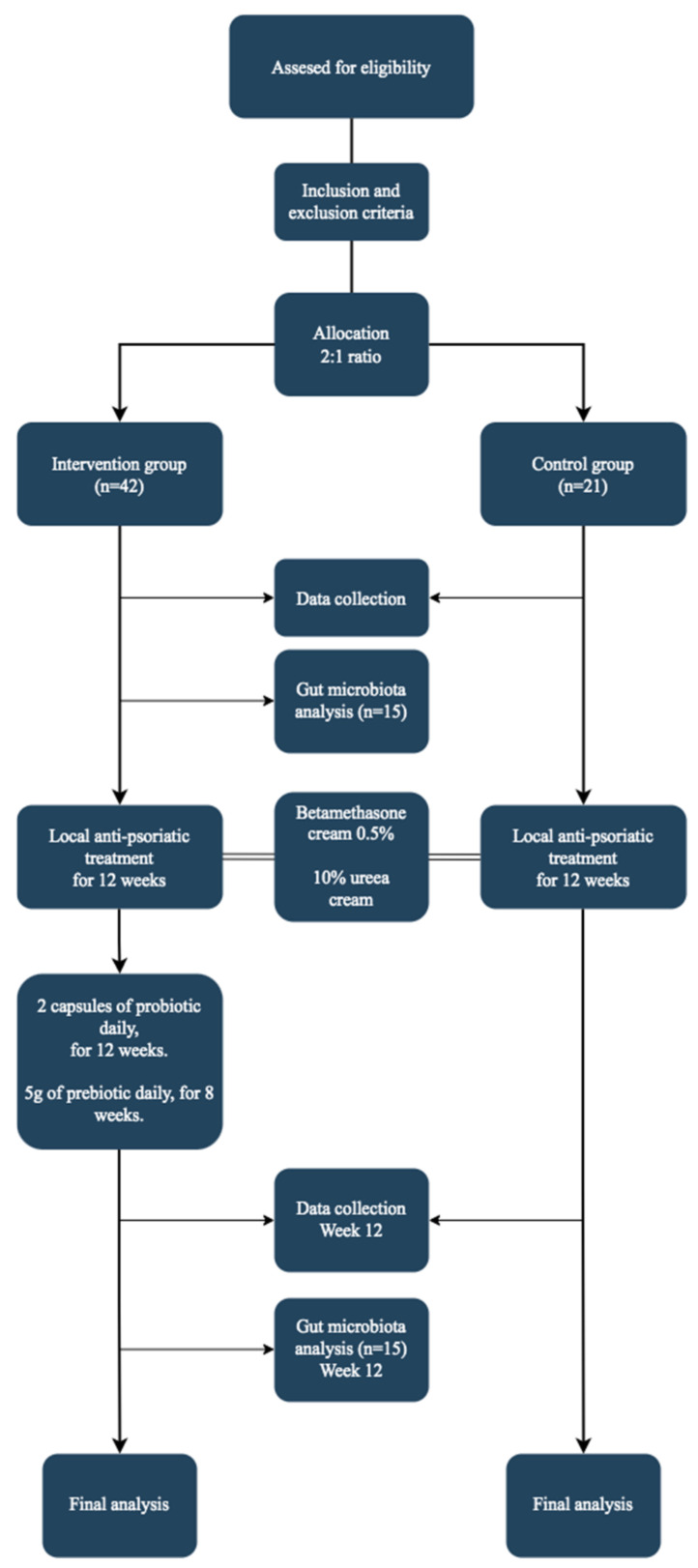
Design and flow of the study.

**Table 1 ijms-24-11225-t001:** Background characteristics.

Variable	Intervention (*n* = 42)	Control (*n* = 21)	*p*-Value
Sex, *n* (%)			
Male Female	18.0 (42.9) 24.0 (57.1)	12.0 (57.1) 9.0 (42.9)	0.29 *
Age (years), mean (SD)			
	34.0 (10.0)	42.9 (7.7)	<0.001 **
Smoking status, *n* (%)			
No Yes	27.0 (64.3) 15.0 (35.7)	15.0 (71.4) 6.0 (28.6)	0.57 *
Family history of psoriasis, *n* (%)			
No Yes	20.0 (47.6) 22.0 (52.4)	8.0 (38.1) 13.0 (61.9)	0.47 *
Psoriasis severity, *n* (%)			
Mild (PASI < 10) Severe (PASI > 10)	20.0 (47.6) 22.0 (52.4)	18.0 (85.7) 3.0 (14.3)	0.006 *
Nail psoriasis, *n* (%)			
No Yes	22.0 (52.4) 20.0 (47.6)	17.0 (81.0) 4.0 (19.0)	0.03 *
Psoriatic arthritis, *n* (%)			
No Yes	24.0 (57.1) 18.0 (42.9)	21.0 (100.0) 0.0	<0.001 *
Inflammatory bowel disease, *n* (%)			
No Yes	17.0 (40.5) 25.0 (59.5)	10.0 (47.6) 11.0 (52.4)	0.59 *
Pruritus, *n* (%)			
No Yes	5.0 (11.9) 37.0 (88.1)	5.0 (23.8) 16.0 (76.2)	0.22 *
Previous corticosteroid treatment, *n* (%)			
No Yes	11.0 (26.2) 31.0 (73.8)	12.0 (57.1) 9.0 (42.9)	0.016 *

* Pearson’s chi-square test. ** Independent-samples *t*-test. PASI, Psoriasis Area and Severity Index; SD, standard deviation.

**Table 2 ijms-24-11225-t002:** DLQI and PASI comparison between the intervention and control groups.

Variable	Intervention (*n* = 42)	Control (*n* = 21)	*p*-Value ***	*p*-Adjusted ^1^ (Change in Intervention Versus Change in Control Group)
PASI, mean (SD)				
Baseline	10.0 (1.8)	8.6 (1.2)		
End of study	8.1 (2.4)	8.7 (1.6)		
Change	1.9 (1.4)	−0.2 (0.6)	<0.001	
*p*-value * (baseline versus end of trial)	<0.001	0.23		
Adjusted mean	7.6	9.8		<0.001
DLQI, mean (SD)				
Baseline	7.2 (2.9)	7.7 (2.9)		
End of study	5.8 (2.9)	8.4 (4.1)		
Change	1.4 (1.0)	−0.7 (2.3)	<0.001	
*p*-value * (baseline versus end of trial)	<0.001	0.19		
Adjusted mean	6.0	7.9		<0.001

* Paired-samples *t*-test. *** Independent-samples *t*-test. ^1^ Analysis of covariance test adjusted for age, sex, and baseline value of the parameter being evaluated. DLQI, Dermatology Life Quality Index; PASI, Psoriasis Area and Severity Index.

**Table 3 ijms-24-11225-t003:** Anthropometric data comparison between the intervention and control groups.

Variable	Intervention (*n* = 42)	Control (*n* = 21)	*p*-Value **** (Change in Intervention versus Change in Control Group)
BMI (kg/m^2^), median (IQR)			
Baseline	24.5 (7.9)	21.6 (4.1)	
End of study	24.5 (8.4)	21.8 (4.0)	
Change	0.2 (0.6)	−0.2 (1.0)	0.025
*p*-value ** (baseline versus end of trial)	0.019	0.17	
Fat mass (%), median (IQR)			
Baseline	29.4 (19.4)	22.7 (5.2)	
End of study	26.4 (22.6)	22.4 (5.4)	
Change	0.4 (2.5)	−0.1 (1.7)	0.32
*p*-value ** (baseline versus end of trial)	0.22	0.92	
Free fat mass (%), mean (SD)			
Baseline	52.6 (10.3)	46.3 (9.0)	
End of study	50.3 (10.2)	46.7 (9.1)	
Change	2.4 (5.4)	−0.4 (1.6)	0.004
*p*-value * (baseline versus end of trial)	0.007	0.27	

* Paired-samples *t*-test. ** Related-samples Wilcoxon signed rank test. **** Independent-samples median test. BMI, body mass index; IQR, interquartile range; SD, standard deviation.

**Table 4 ijms-24-11225-t004:** Inflammatory marker comparison between the intervention and control groups.

Variable	Intervention (*n* = 42)	Control (*n* = 21)	*p*-Value (Change in Intervention versus Change in Control Group)
TNF-α (pg/mL), median (IQR)			
Baseline	50.0 (41.4)	7.3 (54.7)	
End of study	15.1 (36.2)	11.0 (12.7)	
Change	19.5 (42.0)	–1.5 (22.8)	0.040 ****
*p*-value ** (baseline versus end of trial)	0.002	0.82	
IL-17A (pg/mL), median (IQR)			
Baseline	19.0 (89.5)	3.5 (6.6)	
End of study	27.9 (51.8)	12.5 (80.4)	
Change	–1.2 (46.3)	–3.8 (16.9)	0.66 ****
*p*-value ** (baseline versus end of trial)	0.55	0.006	
IL-6 (pg/mL), median (IQR)			
Baseline	21.0 (20.5)	3.8 (1.2)	
End of study	16.2 (16.6)	21.3 (17.8)	
Change	3.1 (10.0)	–2.4 (18.4)	0.002 ****
*p*-value ** (baseline versus end of trial)	0.05	0.26	
IFN-γ (pg/mL), median (IQR)			
Baseline	7.4 (21.3)	5.3 (9.8)	
End of study	5.9 (16.2)	6.0 (4.0)	
Change	1.3 (6.7)	−0.3 (5.8)	0.040 ****
*p*-value ** (baseline versus end of trial)	0.14	0.27	
IL-10 (pg/mL), median (IQR)			
Baseline	4.4 (3.6)	4.4 (11.8)	
End of study	9.0 (16.2)	3.2 (4.3)	
Change	–3.6 (5.4)	0.2 (3.7)	<0.001 ****
*p*-value ** (baseline versus end of trial)	<0.001	0.005	

** Related-samples Wilcoxon signed rank test. **** Independent-samples median test. IFN, interferon; IL, interleukin; IQR, interquartile range; TNF, tumor necrosis factor.

**Table 5 ijms-24-11225-t005:** Metabolic parameter comparison between groups.

Parameter	Intervention (*n* = 42)	Control (*n* = 21)	*p*-Value (Change in Intervention versus Change in Control Group)	*p*-Adjusted
Total cholesterol (mg/dL), mean (SD)				
Baseline	195.5 (31.8)	190.5 (64.1)		
End of study	192.8 (29.0)	203.8 (61.5)		
Change	2.7 (14.0)	–3.3 (43.5)	0.12 ***	
*p*-value * (baseline versus end of trial)	0.23	0.11		
Adjusted mean	188.7	211.9		0.003
LDL-C (mg/dL), mean (SD)				
Baseline	126.4 (28.4)	135.0 (137.2)		
End of study	119.4 (29.6)	149.6 (132)	0.001 ***	
Change	7.0 (15.0)	–14.6 (36.4)		
*p*-value * (baseline versus end of trial)	0.004	0.08		
Adjusted mean	119.5	149.3		<0.001
HDL-C (mg/dL), mean (SD)				
Baseline	52.7 (9.0)	58.4 (11.2)		
End of study	56.6 (10.7)	58.6 (13.6)		
Change	–3.9 (5.4)	−0.3 (8)	0.07 ***	
*p*-value * (baseline versus end of trial)	<0.001	0.88		
Adjusted mean	58.8	54.2		0.027
Triglyceride (mg/dL), median (IQR)				
Baseline	83.0 (42.0)	62.0 (70.5)		
End of study	75.5 (47.0)	65.0 (95.0)		
Change	5.5 (15.0)	–2.0 (19.0)	0.024 ****	
*p*-value ** (baseline versus end of trial)	0.017	0.36		
Uric acid (mg/dL), median (IQR)				
Baseline	4.5 (1.6)	3.6 (3.1)		
End of study	3.6 (0.9)	4.4 (3.4)		
Change	0.6 (0.8)	−0.2 (4.3)	0.32 ****	
*p*-value ** (baseline versus end of trial)	0.001	0.64		
Blood sugar (mmol/L), median (IQR)				
Baseline	90.0 (19.0)	79.0 (46.0)		
End of study	100.0 (24.0)	82.0 (20.0)		
Change	–7.5 (21.0)	–2.0 (29.0)	0.53 ****	
*p*-value ** (baseline versus end of trial)	0.001	0.16		
Insulin (mcU/mL), median (IQR)				
Baseline	17.1 (17.5)	14.3 (22.5)		
End of study	25.7 (18.8)	15.2 (13.0)		
Change	–2.9 (9.4)	−0.9 (6.0)	0.42 ****	
*p*-value ** (baseline versus end of trial)	0.001	0.66		

* Paired-samples *t*-test. ** Related-samples Wilcoxon signed rank test. *** Independent-samples *t*-test. **** Independent-samples median test. C, cholesterol; HDL, high-density lipoprotein; IQR, interquartile range; LDL, low-density lipoprotein.

**Table 6 ijms-24-11225-t006:** Changes in skin ultrasound markers.

Parameter	Intervention (*n* = 42)	Control (*n* = 21)	*p*-Value (Change in Intervention versus Change in Control Group)
Psoriasis plaque hydration (μS), median (IQR)			
Baseline	20.1 (13.4)	61.0 (19.0)	
End of trial	90.0 (31.5)	74.0 (14.0)	
Change	–60.1 (30.8)	–18.0 (23.0)	<0.001 **
*p*-value * (baseline versus end of trial)	<0.001	<0.001	
Perilesional area hydration (μS), median (IQR)			
Baseline	131.0 (74.5)	134.0 (23.0)	
End of trial	168.0 (90.0)	157.0 (38.0)	
Change	–52.1 (30.5)	–22.0 (21.0)	0.003 **
*p*-value * (baseline versus end of trial)	<0.001	<0.001	
SLEB (µm), median (IQR)			
Baseline	261.5 (95.0)	224.0 (100.0)	
End of trial	159.0 (53.0)	186.0 (80.0)	
Change	109.0 (91.0)	40.0 (72.0)	<0.001 **
*p*-value * (baseline versus end of trial)	<0.001	0.009	
Skin thickness (µm), median (IQR)			
Baseline	1168.0 (521.0)	1245.0 (1399.0)	
End of trial	997.0 (150.0)	1302.0 (506.0)	
Change	237.5 (276.0)	20.0 (167.0)	<0.001 **
*p*-value * (baseline versus end of trial)	<0.001	0.82	

* Related-samples Wilcoxon signed rank test. ** Independent-samples median test. IQR, interquartile range; SLEB, subepidermal low-echogenic band.

**Table 7 ijms-24-11225-t007:** Key changes in normally distributed parameters of gut microbiota.

Parameter	Baseline Mean (SD)	End of Study Mean (SD)	Change Mean (SD)	*p*-Value *
Shannon index	2.820 (0.317)	2.990 (0.291)	−0.169 (0.222)	0.011
*Firmicutes*/*Bacteroidetes* ratio	1.206 (0.433)	1.620 (0.334)	−0.413 (0.294)	<0.001
Acetate/propionate production	15.660 (9.023)	20.346 (10.035)	–4.686 (6.531)	0.015
*Prevotella*/*Bacteroidetes* ratio	3.200 (1.107)	2.280 (1.435)	0.920 (1.425)	0.025
LPS-positive bacteria	3.465 (1.817)	2.421 (1.310)	1.044 (1.531)	0.019
*Bacteroidetes*	40.572 (8.160)	30.097 (5.168)	10.475 (6.499)	<0.001
*Prevotella* spp.	19.660 (12.244)	10.523 (7.951)	9.136 (10.118)	0.004
*Prevotella copri*	19.028 (12.358)	8.488 (7.532)	10.540 (9.934)	0.001
*Ruminococcus* spp.	2.479 (1.081)	3.446 (1.497)	−0.967 (1.688)	0.044

* Paired-samples *t*-test. LPS, lipopolysaccharide; SD, standard deviation.

**Table 8 ijms-24-11225-t008:** Key changes in non-normally distributed parameters of gut microbiota.

Parameter	Baseline Median (IQR)	End of Study Median (IQR)	Change Median (IQR)	*p*-Value *
*Verrucomicrobia*	0.543 (1.451)	0.592 (2.115)	−0.199 (0.491)	0.031
*Akkermansia muciniphila*	0.051 (0.526)	0.189 (0.678)	−0.068 (0.238)	0.030
*Clostridium difficile*	0.006 (0.011)	0.002 (0.006)	0.004 (0.006)	0.040

* Related-samples Wilcoxon signed rank test. IQR, interquartile range

## Data Availability

The data presented in this study are available on request from the corresponding author. The data are not publicly available due to privacy and ethical considerations.

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
