# Peer review of "Transforming Psoriasis Care: Probiotics and Prebiotics as Novel Therapeutic Approaches"

_ijms, 2023, doi:10.3390/ijms241311225_

Round 1
Reviewer 1 Report
The manuscript "Transforming Psoriasis Care:Probiotics and Prebiotics as novel Therapeutic Approches" describes a clinical study where a probiotic product consisting of strains of Bacillus and prebiotic were evaluated for there effectiveness in treating psoriasis symptoms. Overall the outcome of the study was positive with study participants that received the probiotics and prebiotic suppelemntation having improved quality of life. Positive changes in the cytokine response, metabolic markers and the host microbiome are also reported. The study appears to be well conducted although not double blinded. The authors thoroughly compare their finding to previously published research and discuss the consistencies and inconsistencies observed. The manuscript would benefit from inclusion of some details on the rationale for the selection of the strains that were included in the probiotic supplementation. Is there preclinical data available for each of the strains? Are the 5 strains necessary? Also the prebiotic component includes 3 polysaccharides, are all 3 necessary? Are the prebiotics metabolised by the bacillus strains?
Author Response
We would like to thank the reviewers for their suggestions, comments and recommendations. We agreed with the observations regarding our paper, and we revised the manuscript in order to improve the presentation of our work. Our responses are given in a point-by-point manner below.
Sincerely,
Dr. Laura Ioana GavrilaÈ™

Reviewer 2 Report
The reviewer believes that this manuscript has flaws; this concerns the design of the study.
It seems that the authors should have applied the correct methods of randomizing patients. Was it correct to randomize patients based on the sequence of their serial numbers? There are other, more adequate methods of randomization. Apparently, due to the fact that the authors used an incorrect randomization method, there are such significant differences between patients in the control group and the intervention group in those indicators that are important for assessing the effectiveness of the therapy proposed by the authors; this concerns significant intergroup differences in the severity of psoriasis, the presence of nail psoriasis, and psoriatic arthritis. Significant intergroup differences exist for initial concentrations of inflammatory markers. In addition, the authors write that there was a proportion of patients taking corticosteroids before the study, but in the Materials and Methods section, it is indicated that corticosteroid use is an exclusion criterion. All this shows that the randomization of patients into groups is incorrect.
In the intervention group, in addition to the probiotic, patients were also prescribed a prebiotic. Obviously, due to the high antagonistic activity (bacilli secrete a variety of antimicrobial substances, often with a wide spectrum of action), probiotic microorganisms can effectively destroy various bacteria living in the intestine, including those that are classified as normal intestinal microbiota. The addition of a prebiotic may allow the normal gut microbiota to proliferate effectively. But here the question arises: how to understand whether the normalization of the intestinal microbiota is the effect of a probiotic or a prebiotic? Or is it a cumulative effect of both drugs? It seems that to answer these questions, the authors should have used additional intervention groups, where a probiotic or prebiotic would be used separately for therapy. And yes, why a probiotic containing Bacillus bacteria? These are bacteria that are not typical for the human intestine. Why not use any other probiotic?
And most importantly, why weren't the same research methods and procedures used for the control group as for the intervention group? For example, it is completely unclear from the manuscript what changes there are in the gut microbiota in the control group. The existence of such data would possibly explain the multidirectional changes in the studied indicators in the groups.
The "Introduction" section does not provide an understanding of the purpose of the study, especially since a significant number of similar studies have already been published previously. It seems that the authors should change this section so that it is clear how this study differs from many similar ones.
Section "Results".
It is very difficult to understand the data in the tables in this section; there are values in round brackets, it is not clear what they mean. In some places, this seems to be a percentage, but the reviewer is not completely sure of this. The authors should explain this in the captions to the tables.
The authors write (lines 66-67) that there was a proportion of patients taking corticosteroids before the study, but in the Materials and Methods section, it is indicated that taking corticosteroids is an exclusion criterion.
It is not clear why the authors provide anthropometric data in the manuscript, which are not discussed later.
Why do groups differ so much in baseline cytokine levels (pre-study)?
Author Response

(The authors gave the same response as above.)

Round 2
Reviewer 2 Report
The authors have made considerable efforts to improve the manuscript, and they have succeeded in doing so. However, the reviewer believes that this manuscript has a slight flaw. This reviewer's remark concerns the abstract of the manuscript. The reviewer believes that the abstract should indicate the possible limitations of this study, which the authors write about in the text of the manuscript in subsection 2.7. Without specifying the limitations of the study, the abstract will mislead readers into drawing incorrect conclusions about the study.
Author Response
Please find attached a revised version of our manuscript “Transforming Psoriasis Care: Probiotics and Prebiotics as Novel Therapeutic Approaches”, which we would like to submit for publication as an Article in International Journal of Molecular Sciences.
The comments of the reviewer 2 were highly insightful and enabled us to greatly improve the quality of our manuscript. By addressing these aspects, we have further strengthened the overall quality of the article, and we hope that now it is ready for publication. All changes were made using track changes in MS Word.
